# Grounding the Conscience

**Angus John Louis Menuge**

Department of Philosophy, Concordia University Wisconsin, Mequon, WI 53097, USA; angus.menuge@cuw.edu

**Abstract:** Most people rely on their conscience as a source of moral intuitions needed to test ethical proposals. Assume that the conscience can deliver knowledge of moral obligations under the right conditions. What ontological resources are needed to explain such a faculty? That depends on (1) the nature of moral obligations, and (2) what it takes to be receptive to them. I argue that close attention to (1) and (2) shows that materialism cannot account for the conscience, but that Christian theism plausibly provides the requisite resources. This is because moral obligations are naturally received as commands, they are prescriptive, personal, and express a kind of universal normative necessity that cannot be grounded in the local contingencies of a materialist world. Moral obligations are expressed as commands of practical reason, and they are knowable only if the world is governed by a divine personal Logos, and there are "Logos beings", beings like God in their receptivity to these commands. Moral obligations are themselves immaterial entities and only a being with an immaterial dimension of the right sort can be receptive to them. This argument parallels a version of the argument from reason that sees theism as the best explanation of our logical reasoning abilities.

**Keywords:** conscience; meta-ethics; moral realism; moral anti-realism; materialism; Christian theism

## 1. Introduction

The significance of the conscience as a guide to action crucially depends on whether one is a moral realist (who believes that knowledge of moral truths is possible) or a moral anti-realist (who denies this, or else radically redefines moral truth). It is notoriously hard to draw the distinction between these positions with precision because there are many varieties of moral realism, and hence many ways to be a moral anti-realist (Joyce 2021). However, in broad terms, it is fair to say that moral realists claim that there are moral truths that can sometimes be known, while moral anti-realists either deny that moral truths exist, or claim that they are unknowable even if they do exist, or else redefine moral truth in ways that realists reject. An example of the last option is Sharon Street's moral constructivism: Street denies that there are moral facts, but claims that there can still be moral 'truths' defined in terms of the reflective equilibrium we reach based on our evaluative attitudes (Street 2006). Realists typically object that we might reach a reflective equilibrium that genocide was right, but that would not make it true that it was right.

Although there have been several distinct accounts of the conscience (Sorabji 2014), a widely held idea (both historically and today), is that the conscience is a faculty that can sometimes provide knowledge of moral obligations. This is arguably part of the pre-reflective common-sense view of the conscience. Interpreted more philosophically, the idea is that conscience grants access to justified (or warranted) true beliefs grounded in moral reality. This entails that some version of moral realism must be true, and that is the assumption of this paper.

In what follows, I will first try to clarify what the faculty of conscience is, and make it plausible that it is capable of providing moral knowledge (Section 2). Then, I will consider the ontological resources needed to explain such a faculty. This will require a consideration of (1) the nature of moral obligations, and (2) what it takes to be receptive to them (Section 3). Next, I argue that close attention to (1) and (2) reveals that materialism cannot account for the conscience (Section 4). This is because moral obligations have inherent semantic

meaning (they are naturally received as instructions or commands), they are prescriptive and personal, and they express a kind of normative necessity that cannot be grounded in the local contingencies of a materialist world. Rather, as I argue in Section 5, the conscience is better explained by Christian theism. This, I maintain, is because moral obligations are naturally received as commands of practical reason that are knowable only if a divine personal Logos governs the world, and there are "Logos beings", beings that are like God in their sensitivity to these commands. I argue that moral obligations are themselves immaterial entities and that only a being with an immaterial dimension of the right sort can be receptive to them. This argument parallels a version of the argument from (theoretical) reason that sees some version of theism as the best account of our ability to ground our thinking in logical principles (Plantinga 1993, 2000, 2011; Reppert 2003, 2009).

## 2. The Conscience and Moral Knowledge

Just what is meant by the word 'conscience'? As Richard Sorabji notes, in its root meaning, tracing to a direct Latin translation of Greek sources, conscience is a form of self-knowledge (Sorabji 2014, p. 14). Citing Sorabji, Joseph Dunne notes that "in Latin: con + scientia = conscientia (or sharing knowledge with oneself)" (Dunne 2021, p. 4). Through conscience, one becomes aware (knows) that one believes that what one has done, or plans to do, is either right or wrong. This knowledge is a second-order knowledge of one's own conduct in light of those beliefs. Conscience is fallible for a couple of reasons. First, a second-order error may occur if one falsely believes that one has a first-order belief about what is right or wrong that one does not have (e.g., if one strongly feels that one ought to believe something and thereby deceives oneself into thinking that one does, as when someone says, "I thought I was a patriot, but I was wrong."). Second, even when conscience does give an agent access to self-knowledge (in the sense of knowledge of what one believes is right or wrong in one's actions), it only provides moral knowledge if there is no first-order error, if those first-order beliefs are true (and are produced in the right way, "justified", etc.). So, our conscience can easily lead us to err. For example, after the Second World War, a small minority of the Nazi doctors retained a clear conscience about their human subject experiments, as they continued to hold the false belief that their actions were morally justified. Likewise, children often suffer a guilty conscience about their parents' divorce, even though they are not to blame.

Thus, the ability of the conscience to provide moral knowledge depends on whether the conscience is properly informed. Did the conscience develop, and was it trained, in such a way that it was oriented to moral truth? And did it have access to a source of moral truth and a reliable means of acquiring that truth from that source? Was it able to detect and reject sources of moral error, "narratives" that might take the conscience captive, rationalizing a wrong action as right, or misrepresenting a right action as wrong? Here, Naselli and Crowley note that the conscience can be damaged so that it is either insensitive or oversensitive (Naselli and Crowley 2016, p. 29). It can become insensitive if we learn to ignore its voice. A contemporary example is that some embrace a political narrative with such fervor that it delegitimizes and dehumanizes those who hold other views so that their pain and suffering when their rights are violated is dismissed or rationalized away ("they deserved it"). The conscience can also become oversensitive, because we impose burdens on ourselves that are not grounded in moral obligations, for example, when a wife punishes herself, wondering what she can do to atone for her husband's substance abuse, as if she were responsible. In this view, the conscience is a bit like a garden, where the voices of true moral obligations must be properly nourished and kept alive, but the voices of false moral obligations must be weeded out. This might help to explain why the conscience is fallible: it is easy to develop a poorly informed conscience.

Nonetheless, as noted, a widely held view of the conscience is that, under the right conditions, it can provide knowledge of moral obligations. Combined with the idea that conscience can deliver a form of self-knowledge about one's own beliefs, this means that, through the right use of conscience, one can acquire beliefs about the morality of one's own

actions that are reliably grounded in moral reality. We might call this double achievement—self-knowledge in light of moral knowledge—moral self-knowledge. Why, though, should one believe that moral-self-knowledge is possible, that such a robust form of moral realism is true? While a full answer is beyond the scope of this paper, I will offer a few arguments that make plausible the existence of moral self-knowledge.

As Thomas Nagel points out, even if moral anti-realism is not directly self-contradictory, it quickly generates paradoxes (Nagel 1997, pp. 101–25). For suppose I believe that there are no objective moral obligations or that I am unable to know them, so that practical reason is never the explanation for my actions: I never do anything (or refrain from doing anything) because I become aware of a binding moral reason to do so (or not to do so). The alternative, apparently, is that I am moved either by non-rational forces (e.g., instincts, emotions, conditioned reflexes) or by non-moral reasons (e.g., prudence, practicality). But suppose further that I come to believe this of myself, that I come to think that I am entirely subject to non-moral causes. I still find that I (at least appear to) have a decision to make, and ask myself, "What is the right thing to do?" Nagel observes that this is what creates the space for reason to operate:

> This step back, this opening of a slight space between inclination and decision, is the condition that permits the operation of reason with respect to belief as well as with respect to action, and that poses the demand for generalizable justification . . . . It is only when, instead of simply being pushed along by impressions, memories, impulses, desires, or whatever, one stops to ask "What should I do?" or "What should I believe?" that reasoning becomes possible—and having become possible, becomes necessary. (Nagel 1997, p. 109)

As Nagel notes, when we ask these normative questions about what we should believe or do, we are not looking for a purely first personal answer, but for a justification. We see this very clearly in how we hold each other accountable. If the police ask a man why he thinks it permissible to be a mass murderer, we cannot understand the reply: "Some people like stamps; I like to kill people." We do not see a preference for killing as any kind of justification for killing, unlike appeals to self-defense, for example. We are asking for "an assessment that should enable anyone else to see what is the right thing for you to do against that background" (Nagel 1997, p. 110).

Nagel's account assumes that we are capable of a kind of double self-transcendence that fits nicely with our account of conscience. First, we are capable of seeing ourselves objectively, as others can see us: we are capable of getting beyond the purely subjective perceptions of and motivations for our actions so that we can see those actions as if from outside ourselves. Second, when we are called to account for our actions, we understand that the reason that we give must also be objective: it must be a reason why anyone so placed should act in this way. This is a request for moral self-knowledge, and helps to explain why a person can say, "When I realized what I had done, I could not live with myself." But are there plausible examples of people acquiring this doubly self-transcendent moral self-knowledge? There surely are.

Consider just one example, which could easily be multiplied. During the early twentieth century, eugenics became popular, with proponents in many countries. In 1930s Germany, the Nazis favored a policy of "national hygiene." Just as medicine aims to keep individual bodies free of dangerous elements (e.g., cancer cells), it was thought that a nation should purify itself of those elements that make it physically or psychologically unhealthy. This led to the Aktion-T4 child euthanasia program targeting "defective" children. As Robert Jay Lifton argues, the moral evil of this program was hidden from its participants because they accepted a narrative presented in impersonal medical terms that concealed the moral evils done to concrete individuals:

> In the entire sequence—from the reporting of cases by midwives or doctors, to the supervision of such reporting by institutional heads, to expert opinion rendered by professional consultants, to coordination of the marked forms by Health Ministry officials, to the appearance of the child at the Reich Committee

institution for killing—there was at no point a sense of personal responsibility for, or even involvement in, the murder of another human being. Each person could feel like no more than a small cog in a vast, officially sanctioned, medical machine. (Lifton 1986, p. 55)

Yet, after the Second World War, when these doctors were interviewed, all but a minority recognized, with horror, that they had been complicit in evil. While it may not be a knock-down refutation of moral anti-realism, it seems that the most plausible explanation of this transition is that the doctors achieved moral self-knowledge. They were able to gain a more objective picture of their actions—as the deliberate killing of innocent children—and became aware that they themselves knew that these actions could not be justified. (Repentant slave-holders, drug dealers, human traffickers, and corporate raiders provide many similar examples.)

This moral realist account is more plausible than the claim that the anti-realist seems compelled to make, that the new perspective of the doctors simply reflects a change in their non-moral reasons, and their non-rational motives and perceptions. One reason to prefer moral realism here is our phenomenological awareness of the difference between existing as a passive spectator of our non-moral motives and experiences, and existing as one who actively takes moral responsibility. As a paradigm case of the former mode of existence, consider Albert Camus's character Meursault in his 1942 work L'Etranger, published in English as The Stranger or The Outsider (Camus 2013). When Meursault kills an Arab on a beach, he is aware of no reason for the action and passively reports his doing it as if he were describing a non-rational process, like oxidation or glaciation. Somehow, the light and heat led him to shoot the Arab. In a sense, Meursault is the ultimate example of a person living as if moral anti-realism were true. Yet, it seems a problem for anti-realism that we do not have to live that way and that we frequently are aware of reasons that may excuse or accuse us. We are all aware of how different our existence is when we take moral responsibility for it. I can become aware of my sloth, my passive acquiescence in practices I should oppose, or as one hymn has it, my "weak resignation to the evils we deplore" (Fosdick 1985). And when I do, I can no longer comfortably go with the flow, and simply watch myself being complicit with evil. Yet it is not just that I feel uncomfortable, which the anti-realist can allow. I also know that I have a decision to make, one which demands a justification in terms of what should be done, not a mere description of what occurs.

Nagel (1997, p. 117) discusses a masterful example from Kant (2015, pp. 27, 124). Suppose that a tyrannical prince tells me that he will kill me unless I bear false witness against a man I know to be innocent. There is, of course, the problem of courage. But whether I am brave enough or not to refuse the prince's command, I am still aware that, whatever its cost to me, obeying the prince is wrong. I cannot simply watch myself sending an innocent man to certain death as if it were not an evil for which I am accountable. If I am a coward, I may do this terrible thing. But I know I will not be able to live with myself if I do. What makes this moral self-knowledge possible is that "I find within myself the universal standards that enable me to get outside of myself . . . I know that I can refuse . . . because I know that I ought to refuse" (Nagel 1997, p. 117).

Thomas Reid argued that humans begin their inquiries with a certain stock of original natural judgments (such as that we exist and endure over time, can come to know the world around us, and can tell the difference between good and evil). If Reid is correct, we might argue that examples of the apparent acquisition of moral self-knowledge like those above reflect the common, original judgment of human beings that moral self-knowledge is possible. The burden of proof therefore lies with the moral anti-realist to show that it is not. Though I cannot argue for this in detail here, I agree with Nagel that the anti-realist cannot meet that burden. No facts about non-moral processes going on in and around us seem able to unseat the intuition that we still have decisions to make that demand reasons for our actions. These reasons seek to justify the commands of practical reason (One ought to do A! One ought not to do B!! etc.), by providing the normative grounds for these commands (why we ought do A or why we ought not to do B, etc.). So, I will assume going forward

that moral obligations are naturally received as commands of practical reason, that these commands are capable of being justified by moral reasons, and that we can sometimes know these commands are morally justified.

## 3. The Nature of Moral Obligations

In this section, I will argue that moral obligations have four leading characteristics that a comprehensive worldview needs to explain. They are naturally received as commands, they are prescriptive and personal, and they possess a kind of universal normative necessity.

### 3.1. Moral Obligations Are Naturally Received as Commands

It is arguable whether or not moral obligations are themselves commands: perhaps moral obligations are direct implications of right-making or wrong-making states of affairs. On this view, if kind treatment makes a state of affairs right, it implies an obligation to be kind, and if cruel treatment makes a state of affairs wrong, it implies an obligation not to be cruel. But even if moral obligations are not commands, they are certainly naturally received by agents as commands, imperatives of the form "One ought to do A!" or "One ought not to do B!", etc. Commands appear to have inherent semantic meaning. Thus, there is a clear difference in meaning between "Do not steal!" and "Do not kill!" A command is an imperative, a species of instruction, and while it is controversial in the philosophy of language whether commands can be understood as propositions (since strict imperatives are not, like declaratives, true or false), it is clear that they are semantically meaningful expressions. This means that, as the study of the theoretical justification of moral imperatives, ethics has a peculiar subject matter. It is very different from say, basic chemistry, in which the fact that chemical formulas are expressed in meaningful terms (e.g., "Calcium sulfate" or "CaSO4") does not establish that chemicals (atoms and molecules) themselves have inherent meaning. In ethics, the very subject matter of the discipline (moral imperatives) bears a semantic meaning.

One might object that an individual who has had no training in ethics, and who would have difficulty in articulating the moral grounds of her actions, could still act morally. But that would not show that the moral command she was following were not meaningful, only that she could not explain the justification of that command. Moreover, even if she found it difficult to express the command, when she decided to act, she formed a volition. Yet volitions are naturally understood as self-commands: she mentally told herself to do something. But thought itself has a content (our mind contains representations or propositional attitudes), and some have even argued that underlying our various natural languages, there is language of thought itself (Fodor 1980). Regardless of whether the latter is right, thoughts are certainly meaningful in their own right, and this includes self-commands.

### 3.2. The Prescriptive Character of Moral Obligations

Moral obligations are prescriptive: they tell us what we ought to do or ought not to do. One of the few results of ethics that is not controversial is that it is impossible to reduce prescriptions to descriptions of natural fact. No matter how great the amount of facts considered, natural facts about what human beings in fact do (descriptions) cannot settle the question of what they ought to do (a prescriptive question), unless one adds tendentious metaphysical doctrines, such as hard determinism or psychological egoism that eliminate or drastically reduce the scope of human choice.

To be sure, the descriptive facts make a significant difference to the applicability of moral obligations. One can hardly say that a paralyzed individual is morally obliged to save a drowning person, or that a hospital must save four patients even if its combined resources only allow it so save three. Yet where moral action is possible, the mere facts that someone prefers not to act, or that a society does not approve of the action, or that it would be very expensive, etc. are not sufficient to show that the action is not morally obligatory all the same.

### 3.3. The Personal Character of Moral Obligations

A more profound but more controversial point is that moral obligations seem to have a personal character. That is, when one is morally obliged to do A, the obligation is ultimately to a person and not to something impersonal. Thus when one breaks a promise, the wrong done is not to the promise itself but to the person to whom one made the promise. In that sense, the obligation is "agent-relative": it is because one made this promise to a particular person that the wrong done by breaking the promise is to that person. King David well expresses this personal dimension of moral obligations in his penitential cry to God, "Against you, you only, have I sinned" (Psalm 51: 4).

But we can see the plausibility of the thesis that moral obligations are personal more easily by considering the alternative. Assuming that the Form of the Good is a moral universal, can one really make sense of wronging the Form of the Good? It is not obvious that recognition of the existence of the Form of the Good creates an obligation to do the good, unless one adds that participation (or lack of participation) in the Form of the Good is what makes states of affairs right-making or wrong-making, and that this has moral implications. But even if the latter possibility is correct, it still does not seem that the Form of the Good is the kind of entity that can issue commands to persons: we cannot be receptive to the commands of the Form of the Good, because it is not the kind of being that can give commands. Rather, it seems that only agents can issue commands. Thus if Socrates had become convinced that volcanic vapors at Delphi (and not the authoritative agency of an inspired priestess or a god) had caused the utterance "Know thyself!" it seems he would not, on that ground alone, have a basis for his view that self-examination is morally obligatory (Apology 38a). And even if the Form of the Good did issue commands, it is hard to see what would give these commands their authority. In virtue of what relationship with the Form of the Good is an agent obligated to carry out its commands?

It seems rather that moral imperatives have the character of a moral law, and that law is personal in a double sense. It is a moral law for persons: we can make no sense of moral commands for rocks or other non-persons that cannot understand the law or choose to apply it to themselves. But it also seems to be a moral law from and to persons. Thus, I can understand the idea of a moral command from a governing official to serve in a war, but not the idea that because we are living in dangerous times, I am obliged to fight. Perhaps, given the dangerous times, I have some sort of obligation to defend myself (as Hobbes thought) based on the natural right to self-preservation. But even if this can be cashed out without committing the naturalistic fallacy, this would not ground an obligation to fight for my country since this may not be necessary for self-preservation and actually puts it at risk for the sake of a greater good. Moreover, when it comes to national defense, the official may have the authority to issue that command, but dangerous times (some set of facts about the circumstances) do not have that authority. And if I do not fight when I should, the wrong I do is to persons (not only to the governing official, but to those I am called to defend), but not to the dangerous times.

### 3.4. Moral Obligations Express Universal, Normative Necessity

The agent-relative nature of moral obligations might make one doubt that they could be universal: are not different agents situated differently, and is not that why they have different obligations? But as Nagel realizes, the fact that moral obligations are agent-relative in this sense is compatible with their universality because the demand of practical reason is that one produces a justification for one's actions which would apply equally well to any other agent in the same circumstances. As Nagel points out, this is implicit in the very idea of reason: "a reason is something one person can have [for an action] only if others would also have it if they were in the same circumstances (internal as well as external)" (Nagel 1997, p. 119).

Thus, although moral obligations are agent-relative, the justification of those obligations is agent-neutral, once we fix the circumstances. That is why it is telling when someone says, "A good person would not have done what you just did" or "I would have done

just the same in the circumstances: there were no better options." When we hold people accountable for their actions, the account we seek is not a biographical explanation of why they did those actions, but a justification for anyone so placed to act in the manner that they did.

Moreover, moral obligations have a kind of normative necessity. This is captured by the fact that a positive moral obligation is something we morally must do, and a negative moral obligation something that we morally must not do. Unlike permission, which tell us about possibilities (what we may or may not do), moral obligations tell us of necessities. But these are not descriptive necessities (like the laws of nature), that hold independent of our agency, but prescriptive necessities, as we remain free to follow them or not. What we are told is that morally speaking, we are required to act in a certain way.

Hopefully, at least for those who accept that there are objective moral obligations, this analysis of their character should seem plausible. In the next section, I will argue that materialism does not account either for the existence of moral obligations or for our knowledge of them. Thus, as we have defined it, materialism cannot account for the existence of conscience as a faculty capable of moral self-knowledge.

### 4. The Failure of Materialism to Explain Knowledge of Moral Obligations

*4.1. Moral Obligations Are Not Grounded in a Materialist World*

There are several reasons to think that moral obligations lack a plausible foundation in a materialist world.

First, we saw that moral obligations are naturally received by agents as commands, and these commands appear to be inherently meaningful, because they are a species of instruction. But it is hard to see how instructions can emerge from blind matter. Aristotle taught us that there is a difference between the material cause (what something is made of) and the formal cause (which accounts for a thing's particular form or structure). Moreover, in our experience, materials and instructions belong to discrete ontological categories. This is why we pity the person with a large box of parts from IKEA that fails to include the instructions: the instructions do not emerge from the parts. Two major discoveries of twentieth century science reinforce this conclusion. Life is not simply the result of chemical stuff, but crucially depends on voluminous instructions encoded in DNA (Meyer 2009). A universe compatible with intelligent life is also not something that naturally emerges from the stuff inside the universe (or multiverse), but crucially depends on the fine-tuning of many parameters, including the constants that govern the four fundamental laws of nature: gravitation, electromagnetism, and the weak and strong nuclear forces (Meyer 2021). This fine-tuning consists of non-arbitrary information. If we imagine a universe control center with dials governing the four fundamental forces, then fine-tuning consists of the instructions that specify the correct positioning of the needles on those dials.

The point in both cases is that nothing in matter itself (in living organisms or the universe) implies that the instructions have to exist or have the specific content that they do. This is why material worlds with no life at all, or no intelligent life, are perfectly conceivable: all we have to do is imagine a world with no instructions or significantly different ones. So it seems that information, in the form of instructions, is in general something more than matter itself. The matter does not govern (determine) the instructions, but the instructions do govern the character of matter, and what the matter does. In Aristotelian terms, formal causes cannot be reduced to, and do not plausibly emerge from, material causes. They are something sui generis.

To be sure, a richer form of naturalism, which Stewart Goetz and Charles Taliaferro call "broad naturalism", might allow all of this, as it may grant that nature includes formal causes (Goetz and Taliaferro 2008). However, merely to assert that formal causes are part of nature makes them a brute fact, and so theism is poised to provide a better explanation.

Second, the problem of accounting for the existence of instructions governing reality is more acute for moral obligations, because here the instructions are prescriptive in a strong sense. They do not merely specify how inanimate matter is to be assembled, they tell people

with free will how they should act. This means that in addition to all of the stuff in the world, and the way that stuff is arranged, there is an additional layer of reality consisting of prescriptions that specify how a certain subset of beings—intelligent, rational agents with free will—should act. J. L. Mackie, a naturalist, admitted that the existence of this additional layer of prescriptive reality is not plausible in the godless world of materialism:

> objective intrinsically prescriptive features ... constitute so odd a cluster of qualities and relations that they are most unlikely to have arisen in the ordinary course of events, without an all-powerful god to create them. (Mackie 1983, p. 115)

One reason to accept Mackie's assertion is that in a materialist world there is no basis for teleology. In the materialist world, effects occur because of their blind, unintelligent causes, and of no particular event can one say that it was supposed to happen or supposed not to happen. Analysed from the point of view of materialistic science, the actions of Mother Teresa caring for the poor and of Romanian Communists torturing Pastor Richard Wurmbrand are simply different arrangements of elementary particles. No material entity or force implies that the former action was supposed to happen, and is therefore right, while the latter action was supposed not to happen and is therefore wrong.

> Thus Nietzsche was on strong ground when he concluded that if materialism is true,

> there are altogether no moral facts. Moral judgments agree with religious ones in believing in realities which are no realities. Morality is merely an interpretation of certain phenomena—more precisely, a misinterpretation. (Nietzsche 1977, p. 501)

Modern materialistic science rejects final causes (it denies that anything happens in fulfillment of a purpose, or in order to achieve a goal). But it seems that moral facts cannot be located in a world void of final causes, for without a target (an end) it is senseless to talk of actions either hitting or missing their target, and therefore impossible to say what one ought or ought not to do.

About the most plausible response that a non-theist can make to this problem is the one offered by Erik Wielenberg: godless normative realism (Wielenberg 2014). Wielenberg admits that moral obligations are non-natural facts: they cannot be reduced to any facts about the natural world, as G. E. Moore had argued (Moore 1903). Nonetheless, he claims, certain facts about the natural world make it the case that moral obligations obtain, where "making" is a brute form of necessitation. Thus one action's having the non-moral property of being kind makes it have the moral property of being right, and another action's having the non-moral property of being cruel makes it have the moral property of being wrong.

I have responded in detail to Wielenberg's views elsewhere (Menuge 2019), but here I will just note that his account does not really explain why any moral obligations exist or why they have the particular character they do. For absent any source of teleology, telling us how the world should go, there are conceivable worlds in which all the non-moral facts are the same, but there are no moral obligations, or in which moral obligations are the opposite of what we find them to be in our world. Given the non-moral facts, why should the world not be just as Richard Dawkins describes it?

> The universe that we observe has precisely the properties we should expect if there is, at bottom, no design, no purpose, no evil, no good, nothing but pitiless indifference. (Dawkins 1995, p. 133)

Yet if moral obligations do somehow emerge, why should it not be the case that we ought to be cruel and ought not to be kind? Simply saying that this is not how things worked out seems to rely rather heavily on some form of luck or chance. So, other things being equal, any worldview that has a better explanation for the existence and character of moral obligations has an advantage over Wielenberg's godless normative realism.

Third, a deeper problem for materialism is that it is bound to say that moral obligations ultimately arise from impersonal sources. Materialism seems to have only two options: it

must claim that moral obligations are grounded in material stuff itself, or it must say that moral obligations are grounded in something that emerges from that stuff. The first option seems literally incredible. Can we really make sense of a materialist David crying, "Against stuff, stuff only, have I sinned"? Notice that the intuition remains the same if we replace "stuff" with more sophisticated materialist fillings, like "elementary particles", "forces", "fields", "strings", "membranes", or any other characterization of the universe provided by modern physics. What seems to drive the intuition is that we cannot understand what it means to sin against (or be accountable to) anything impersonal. Yet the second option is also problematic. Suppose my critique of Wielenberg is mistaken, and rightness emerges from kindness just as he says. Still, if I come to believe that "rightness" emerges from kindness in this fashion, it is not obvious that my moral obligation is to rightness itself. That it is wrong to be unkind to people depends ultimately on the value of people. It is (ultimately, at least) not rightness itself that I wrong when I torture Joe, it is Joe. It is true that we talk of violating moral codes, but the authority of the code depends on the value of those whom it protects. So moral obligations do not float free of any beings to which they apply. After all, imagine a possible world in which it is "unkind" to pulverize a rock and this makes it the case that it is "wrong" to do so. Can this create a moral obligation for me to refrain from pulverizing rocks unless rocks are the sorts of things one can wrong, or unless there is some person whose rocks they are whom I can indirectly wrong by damaging his property? For obligations to make sense, moral properties (like rightness and wrongness) must be grounded in a being of a kind that one can treat rightly or wrongly.

If this is right, then the only way for the materialist to overcome this objection is by showing that persons themselves are the sorts of things that emerge from matter. However, that would take us to a family of problems faced by materialist accounts of persons. One problem is the "hard problem of consciousness" (Chalmers 1996, pp. xi–xii): why, given all the physical facts, is there something it is like to be a person? The totality of physical facts does not appear to entail that consciousness exists: it is a further fact. Other problems center on the unity of consciousness, free will, the argument from reason, and the problem of intentionality. These problems have proven to be at least as difficult for materialism to address as the problem of accounting for moral obligations.

Fourth, we saw that moral obligations plausibly have a kind of universal, normative necessity. They imply that if one is obliged to do A, anyone in the same circumstances (internal and external) must do A. Due to their universality and necessity, moral obligations obtain not only in the actual circumstances that have occurred (or that are occurring now), but in future and counterfactual cases as well. This is difficult for materialism to explain, because if moral obligations emerge from the properties of matter, moral obligations must be contingent, for they depend on contingent states of affairs (such as the particular stuff that exists, and the particular events that have occurred in the world's history). Suppose that in the past, matter somehow made it the case that certain actions were obligatory. It would not follow that any future or counterfactual actions are obligatory. For whatever types of events happened in the past need not recur in the future. So if the history of matter changed enough, it might be that different actions would become obligatory. That something has been the case will never show that it will continue to be the case, or that it must be the case. In fact, it seems that the best a materialist can say is that the material history of the world grounds certain moral "rules of thumb", rules that have seemed (perhaps were) obligatory in the past (or now), and which may persist, but which cannot be relied on to continue in the future, and that have not been shown to be necessary (so counterfactual moral thought experiments are unwarranted).

One famous example of the problem will suffice. Charles Darwin thought that human morality was dependent on our natural history, and in particular, on our mode of social organization. From that he concluded (on one reading) that the moral obligation not to kill brothers and female infants might have been different. (On another reading, Darwin thought only that our beliefs about moral obligations might have been different.)

> If . . . men were reared under precisely the same conditions as hive-bees, there
> can hardly be a doubt that our unmarried females would, like the worker-bees,
> think it a sacred duty to kill their brothers, and mothers would strive to kill their
> fertile daughters; and no one would think of interfering. Nevertheless, the bee,
> or any other social animal, would gain in our supposed case . . . some feeling of
> right or wrong, or a conscience. For each individual would have an inward sense
> of possessing certain stronger or enduring instincts, and others less strong or
> enduring . . . . In this case an inward monitor would tell the animal that it would
> have been better to have followed the one impulse rather than the other. The one
> course ought to have been followed, and the other ought not; the one would have
> been right and the other wrong. (Darwin 1998, p. 101)

Yet, the intuition of most people is that if it is wrong to kill brothers and female infants,
no contingent change in human social organization could make it right. We do not believe
that if some statist tyrant, enamored of entomology, made us live like hive-bees, that this
would make it the case that fratricide and female infanticide were right. Likewise, we do
not believe that if "our way of life" changes to include dependence on slavery, that this
could make slavery right. To be sure, contingent changes in the world's material history
would produce different circumstances, so that different moral obligations might apply
(after all, they might lead to the existence of fewer, more, or different people). Yet it does not
seem that the force of moral obligations themselves is dependent on this history. Whether
or not there will be any people (or many, or a few people) in the future, there are things
that we know right now are (and must be) right or wrong to do to people.

One challenge to the idea that universal normative necessity is incompatible with a
modern naturalistic attitude is Christine Korsgaard's voluntaristic reading of Kant. According to Korsgaard, moral values are conferred by the choices of a rational being (Korsgaard
1996a, p. 126). She thinks that Kant's view is that practical reason is something that we
impose on the world, and that we thereby create moral obligations (Korsgaard 1996b,
pp. 4–5). However, this is clearly an anti-realist view, and it is both implausible and a
doubtful reading of Kant. If Korsgaard is right, then the contingent choices of finite beings
somehow bring moral obligations with universal, normative necessity into being. Yet,
this seems no more plausible than the idea that a contingent human choice could create a
new law of logic. The world of rational norms (whether of theoretical or practical reason)
seems rather to be out there, waiting to be discovered. Further, as Michael Rosen (2012)
has pointed out, Korsgaard seems to misread Kant, because Kant clearly believes that
the dignity of human beings precedes any choices that we make. In particular, Kant's
discussion of suicide shows that he thinks suicide could not be made right even if a person
could will that all people committed suicide. Moreover, why are rational beings valuable
in the first place? It cannot be that rational beings are the source of value and that their
choices confer value on themselves (Rosen 2012, pp. 151–53).

### 4.2. Materialism Cannot Account for Our Knowledge of Moral Obligations

Suppose that everything I have said in 4.1 is mistaken and moral obligations can
be grounded in a materialist world. Still, that will not explain the conscience unless
materialism can explain our knowledge of moral obligations. Yet, materialism seems
incapable to explaining how we could acquire moral knowledge. First, as Erik Wielenberg
concedes, moral properties are not natural or material (they are not studied by any of the
natural sciences; they not only cannot be detected by the senses or any scientific instrument,
but also do not serve as theoretical entities in any natural scientific account). Yet, a basic
commitment of materialism is causal closure: there cannot be any non-material causes,
for otherwise the soul, God, etc. could get in. So, as Wielenberg also concedes, moral
properties must be epiphenomenal: they may emerge from matter, but cannot have any
effects of their own.

Wielenberg offers a sophisticated account of how we might come to know moral
properties even though they are epiphenomenal, which I have criticized in detail elsewhere

([Menuge 2019](#)). But the most fundamental problem is this. Before one can have moral knowledge, one must have moral beliefs (true or false), and before one can have moral beliefs one must have moral concepts. Yet, if moral properties are epiphenomenal, they can play no causal role in our acquisition of moral concepts. If so, on purely materialistic grounds, why suppose that any moral concepts we do have are grounded in moral reality? Perhaps, as Nietzsche thought, our ordinary concepts of right and wrong are not derived from the material world, but are merely interpretations of that world. If that is the case, even if there are moral properties out there in the world, our moral concepts could be systematically at variance with them, a work of fiction. Matters are made worse by the fact that, for the materialist, naturalistic evolutionary theory provides a plausible explanation of why we would have the moral concepts that we do even if no moral properties existed. For supposing there are no moral properties (or they are radically different than we suppose), it may still be a fact that cooperation favors our genetic interest and that this explains why we think cooperation is right. Moreover, even if there are moral properties, and even if some of our moral beliefs happen to be true, this is merely a lucky coincidence, because if moral properties are epiphenomenal, we do not have reliable access to them, so these beliefs do not qualify as knowledge.

A second reason for doubting that materialism explains moral knowledge is more fundamental. Moral properties are immaterial entities and therefore cannot stand in any material relation to purely material beings. It is not just that moral properties cannot cause moral concepts: they also cannot stand in spatial, temporal, or any other material relation to a human brain. So there is no physical relation between the brain and a moral property on which moral knowledge could supervene. In order to explain moral knowledge, one must first explain the existence, not of another relation, but of a property or power—intentionality—which allows us to have any kind of knowledge of reality. Intentionality is clearly not a relation, at least in the sense of standard physical relations, since the latter require both relata to exist somewhere at some time, but one can think of objects that never will be exemplified in space and time, like elves and hobbits. Intentionality is better understood as a property or power that allows a sentient being to think about things beyond itself. Yet, intentionality does not seem to be a physical property or power or one that is reducible to physical causation. One can think of future, actually non-existent, and even necessarily non-existent things, but no physical relation to the brain explains this. A future event does not have the physical power to make me think of it, and non-existent objects have no physical (or any other) power at all. Thus, the power of intentionality does not require the object of thought to have the power to make me think of it, but resides in the subject. Moreover, to think about moral obligations requires intentional content, e.g., the thought that fratricide is wrong. Yet, intentional contents also seem to be immaterial entities. We have various propositional attitudes (such as the belief that p, the desire that q, the hope that r), but these propositions (p, q, and r) are intentional contents that cannot be located in a materialist world. They have no coordinates in space-time, cannot be observed or measured, and are not recognized as theoretical entities in any natural scientific theory. If moral obligations, and our thoughts about them, are (or involve) immaterial entities, it is hard to see how materialism can explain moral beliefs, far less moral knowledge.

## 5. Christian Theism Grounds Knowledge of Moral Obligations

Materialism does not seem able to account for either the existence of moral obligations or our knowledge of them, and so cannot account for the faculty of conscience. In this section, we will consider just one competing view, Christian theism, and argue that it provides a much more plausible account of the conscience. In this context, by Christian theism, I mean a version of theism which is informed by both the New and the Old Testaments, and which sees Christ as the foundation of reality. Of course, on some issues, Christian theism will agree with other versions of theism, though I will suggest that the specifically Christian understanding of Christ as logos has some advantages.

### 5.1. The Foundation of Reality Is Logocentric and Teleological

We saw above that moral obligations are naturally received as commands, and these commands have inherent semantic meaning. This is much less surprising on Christian theism than it is on materialism. This is because a major claim of Christian theism is that meaning is not, as in materialism, an emergent phenomenon, with properties radically different from the matter from which it is supposed to have emerged. Rather, meaning (the meaning of God's Word) is foundational to reality: it is what gives things their very being. Thus, at the beginning of John's Gospel, we read, "In the beginning was the Word, and the Word was with God, and the Word was God. He was in the beginning with God. All things were made through him, and without him was not any thing made that was made" (John 1:1–3). Here, "Word" (logos) has a double meaning. It carries the connotations both of Greek philosophy and of Trinitarian theology. From the Greek tradition, logos means (inter alia) the reason why things are as they are. And from Trinitarian theology, the Logos is the second person of the Trinity, the Son, as is made clear in a parallel passage in Paul's letter to the Colossians, which tells us that the one through whom "all things were created" and in whom "all things hold together" is the one who made peace with humankind by "the blood of his cross" (Col. 1:15–20). This understanding of the logos is unique to Christianity (it is not found in Judaism or Islam, for example). It is also noteworthy, that it is because Christ is one person with both a human and a divine nature, that we have an explanation of how divine commands can be expressed in humanly understandable form. Through the communication of the divine and human attributes of Christ, he communicates God's will in a way that we can comprehend. A problem for versions of theism that insist on God's absolute transcendence is that they find it difficult to explain how finite human beings could understand God's will.

Putting these elements together, reality has a foundation that is both semantically meaningful and personal. It is therefore not surprising to see that there exist entities like moral obligations that are inherently meaningful and personal in character. They are not oddities that require a special explanation but particular examples of the ubiquitous structure of reality. If the entire universe exists because it was created by the personal Word of God (Hebrews 11:3), it is much less surprising than it is on materialism that the Creator would generate further commands for those beings with reason and free will, since they can understand those commands and yet they are not compelled to follow them. Human beings are given the terrible dignity of defection: unlike the planets that must follow the courses ordained for them, they are capable of disobedience. So they need a properly informed conscience to deter them from making the wrong choices out of pride and self-love.

This also illustrates the fact that the foundation of reality is teleological: what exists, exists for a reason, for the reasons of the one who made it. All things are made not only through Christ but "for him" (Col. 1:16). In particular, the conscience serves as a communication channel, by which Christ can call us away from enmity to God (since human beings sinfully wish to be their own god) and back to obedience. Without this faculty of conscience, it is unclear how fallen human beings would even be capable of discovering their own sin, or how they could repent. Since God wants all people to be saved and to come to know the truth about themselves and their relation to God (1 Timothy 2:4), it is therefore not surprising that the faculty of conscience is implanted in all human beings (Rom. 2:14–15).

### 5.2. Moral Obligations Are Immaterial Entities

We saw above several reasons for thinking that moral commands, the natural expression of moral obligations, are immaterial entities. First, moral commands are a species of instructions and instructions do not seem to be the sort of thing that emerges from matter: rather they are most naturally understood as belonging to a distinct ontological category that governs matter. Chemicals do not naturally assemble into living things: the right instructions are needed. Forces do not naturally produce intelligent life: they must be finely-tuned by instructions that set their governing constants to the right values. This

immateriality of instructions for assembling and governing matter fits naturally with the Christian Theist assertion that the Word of God precedes the material universe, "so that what is seen was not made out of things that are visible" (Hebrews 11:3). Yet at the same time, this Word can be conveyed in and through material vehicles, whether DNA, the laws of nature, or the spoken sounds or written marks used to express special revelation.

Secondly, moral commands seem to be immaterial because they are prescriptions, but matter does not seem able to tell people what they should do. Ultimately, valid moral commands can only be issued by a rational being with the right authority. On Christian theism (as on other versions of theism), God is a rational spirit, so Christian theism does not face the same difficulty as materialism in explaining how the being could issue an immaterial command for other rational beings. Moreover, the one who created human beings, and who knows what is best for them, seems to be in the best position to tell them how they should live their lives. To be sure, it is important to show that God is good, so that His commands cannot be dismissed as the edicts of an arbitrary tyrant. Yet, it is precisely the position of Christian theism that God is the very standard of goodness (Luke 18:19), and that He is not an impersonal Form of the Good, but a personal loving being who wants the best for those He made specially in His own image (Jer. 29:11, Rom. 8:28). Thus, the moral commands of God, accessed through conscience, are for our good, and when we are not captive to worldly ideas and blinded by sin, we are capable of knowing that (Rom. 12:2).

### 5.3. Moral Obligations Can Be Known Only by a Being with an Immaterial Dimension

For a being to know moral obligations, that being must be receptive both to moral commands and to their justification through practical reason. As we saw, this is hard to square with materialism because: (1) the physical brain does not seem able to acquire moral concepts, (2) obligations are immaterial entities, and (3) obligations have a universal normative necessity that cannot be grounded in the contingencies of the materialist world. Thus, whatever is receptive to moral commands must be something that transcends the limitations of matter, and hence must have an immaterial dimension. That does not by itself entail that the being is entirely immaterial, but that at a minimum, in addition to any material properties the being may have, it also has some immaterial properties. If God is an immaterial being that issues immaterial commands, it is plausible that the receiver of those commands must also have an immaterial dimension.

Yet not just anything immaterial will do. The number 5 and being the successor of zero are both immaterial, but they do not account for knowledge of moral obligations. What specifically is needed is a faculty sensitive to moral prescriptions that can recognize when a prescription fulfills the requirement of universal normative necessity. In a sense, this is a god-like ability, as it requires one to be able to transcend one's own contingent limitations and see what any being so placed must do. This is one place where the Christian teaching that human beings are made in the image of God does real work. To be made in God's image is not to be God, but to be like God in important ways. One way in which we are like God is that we can discern that there are moral obligations necessarily binding on all (created) persons. This requires us to have rational and intentional powers that take us beyond our own particularity and limitations so that we can see others and ourselves objectively, as it were from God's point of view. As Thomas Nagel has frequently emphasized, this same kind of self-transcendence is also required for theoretical reason. For when, for example, one knows that modus ponens is a valid form of inference, one does not merely know that the rule has worked in the past, but knows that it must always hold and that the rational thinker is obliged to follow it wherever it applies.

From this it is not hard to see that the immaterial dimension that accounts for our knowledge of moral (and logical) obligations must be one that contacts universals and the relations between them. That compassionate treatment of the outcast is right and that breaking promises is wrong cannot be known if it is based on personal or even wider human experience, as that will never show that these relations hold of normative necessity. Rather our rational and intentional powers must connect us to the universals (compassion,

the outcast, promises, right, wrong, etc.) so that we discern that these relations must hold in just the same sort of way we discern the relations presented in a law of logic are universal and necessary.

Yet, how is all of this possible? One part of the answer is that our intentional powers enable us to transcend our creaturely limitations in thought. Though being only about six feet tall, I have no difficulty in thinking about the Grand Tetons or the Alps. Though Stephen Hawking spent much of his life confined to a specially designed wheelchair in Cambridge, in his work as a cosmologist, he said, "I have spent my life travelling across the universe, inside my mind" (Hawking 2018, p. 3). So the power of intentionality vastly outstrips the physical limitations of a human being, making it possible to speculate about the origins of the entire cosmos. Moreover, as previously noted, through intentionality, we can think not only of future events, but also of counterfactual ones, and this enables our minds to enter the logical space of all possible worlds. In addition, intentionality leverages the human power of abstraction, which enables us to conceive of universals, including right, wrong, justice, etc. Combining these two abilities, reason enables to "see" that there are relations between universals that hold in all possible worlds. In this way, we discern the laws of logic and the justification for holding that a moral prescription has universal normative necessity. Thus, part of what it means to be made in the image of God is to have the intentional powers necessary to contact the realm of necessary relations between universals.

Secondly, for reason to help us see relations among moral universals, it must be capable of understanding the relationships between the entities with which intentionality acquaints it. After all, an average person acquainted with various mathematical relations often does not understand why they obtain. Likewise, one might be able to think of right, wrong, promise-keeping and betrayal, and be aware that there are relationships between these entities, yet fail to see why these relations must hold. So while reason depends on intentionality (since it concerns what our thoughts are about), it goes beyond it, by providing insight into why relationships obtain between the objects of our thought. What reason contributes beyond intentionality is the power to analyze intentional objects so that the necessity of these relationships becomes apparent. As an analogy, consider a student presented with a right-angled Euclidean triangle for the first time. Through intentionality, the student understands what the right-angled triangle is, and that there is a relationship between the hypotenuse and the two shorter sides. Yet it takes reason (following the proof of the Pythagorean theorem) to see why this relationship must obtain.

It is an old argument, going back at least as far as Plato and Aristotle, that the powers of reason transcend the powers of the material organism: they point to an immaterial soul or at least an immaterial nous (insight, understanding). In its ability to understand necessary relations between universals, reason is not well explained by materialism, which must limit itself to what emerges from the contingent history of matter. Rational powers, like intentional powers, therefore seem to be immaterial powers of a human being, regardless of which more specific anthropology one embraces. They belong to "Logos beings", beings that are like God in their ability to discern the logos that governs all things, including the moral law that is the source of their moral obligations. This is the sense in which human beings, unlike animals, are under the moral law. The moral law is a moral law for us only because we can discern the relationships between universals needed to understand the moral law.

We must also be agents, having the free will to obey or disobey the moral law. We cannot claim to know that we have a moral obligation to do something unless we are justified in believing that we have a choice. For example, it makes no sense to say that a person with an inherited defect that (so far as he knows) cannot be remedied has an obligation to remove the defect. Divine determinists aside, most Christian theists claim that we are like God in having the power of self-determination, hence an ability to obey or disobey moral commands. This does not seem to be a material power, as matter seems to be passive, doing what it does because previous events determine or fix the chances of

its doing so. To be sure, compatibilists claim that in some sense, free will is compatible with this kind of determinism. But as I have argued elsewhere, compatibilism does not seem to be consistent with the existence of acts of reasoning, even if it is consistent with the existence of non-agents like computers that are designed to operate in accordance with reason (Menuge 2011). Yet the conscience, as we have understood it, makes no sense unless it is up to us to follow the commands of practical reason. If that is right, then the conscience requires libertarian free will, which is not compatible with materialism. Moreover, if free will transcends the powers of matter, and knowledge of our free will (the choice is up to us) is one of the things required for us to know we have moral obligations, this is another reason to think we have an immaterial dimension.

Thus, the powers of intentionality, reason, and free will are all required for us to know we have moral obligations, and since these powers appear to be immaterial, we have good reason to conclude that only a being with an immaterial dimension of the right sort (one with these powers) can know it has moral obligations. These powers are much less surprising in Christian theism than in materialism because on the former view (unlike on the latter), they did not emerge from entities that lack these powers, since God Himself has intentionality, reason, and free will and beings made in God's image inherit similar (though more limited) powers.

*5.4. Ultimately, Conscience Involves Interpersonal Relations*

We argued that moral obligations are ultimately to persons. This is true even of our treatment of non-persons. If I steal your favorite bobblehead, it is you I wrong, not the bobblehead. And if I shoot your dog, I arguably wrong you more than the dog. Further, to the extent that I do wrong the dog (and I agree that I do), Christian theism (in common with other versions of theism) maintains that this is because the dog is God's creation and that I was not authorized to destroy it for no reason. So when we wrong non-persons, it is at least arguable that this is ultimately because there is a person whom is wronged. Certainly, while it is wrong to wantonly kill a dog that no human owns (since God made the dog "good", with intrinsic value), one is not accountable to this or any other dog for the action. It seems rather that we are morally accountable only to persons. Even non-theists who deny all this are likely to grant that many moral obligations find their ground in the value and dignity of persons. So how do we explain this fact?

If all (or at least many) moral obligations hold in virtue of personal relations, it must be that persons have a claim on us. It is wrong for me to break a promise to Joe because Joe is the kind of being that can be wronged and he has a claim on me to do right by him. This makes sense in Christian theism because Joe is an image bearer, giving him a special moral value, and because I am an image bearer, which equips me to understand my moral obligations to other image bearers (and indirectly to their possessions, reputations, etc.). But I can also see that the moral obligation to love my neighbor is valid not merely by an analysis of universals (our previous point) but because I can see it issues from an authoritative source. God is the one who created both my neighbor and me and He knows best how I should treat my neighbor. When God tells me how to treat my neighbor, it is as if the designer of the IKEA furniture himself tells me how to put it together.

Further, scripture is clear that whenever we wrong our neighbor, we wrong God. It is wrong for me to kill my neighbor like livestock because God created my neighbor in His image (Gen. 9:6), and I dishonor God when I harm those like Him. Likewise, we are found hypocritical when we bless God but curse our brother made in the image of God (James 3:9), or claim to love God when we hate our brother (1 John 4:20). Certainly, we wrong our brother in doing this, but we also wrong God because we treat those beings God has accorded a special status as if He had not done so, and we claim a superiority to them that God alone enjoys. That, I think, is why David insists that, although he obviously wronged Uriah, in an ultimate sense it was God only that he had sinned against (Psalm 51:4). That all wrongs, both to persons and (arguably) non-persons, ultimately involve wrong to a person

makes sense if all created things are God's property (Psalm 24:1), and we are accountable to God for our treatment of it.

This interpersonal character of moral obligations also helps to explain a widely noted feature of conscience. From Plato's Socrates to Martin Heidegger, conscience has been described as a voice, or a call. When Socrates was told by the Council of Thirty (a vicious oligarchy) to bring in Leon of Salamis for execution, his conscience conveyed the voice of "the god," who told him he must not obey. Likewise, people often say things like "My conscience called me to do it." So conscience seems to involve a meaningful message—a voice or call—and the source of a voice or call arguably must be a person. True, we say that the potato chips are calling us to eat them, but we do not mean it literally. In fact, we are concocting a cover story, a spurious call, to rationalize our behavior. We are, without appropriate authority, trying to call ourselves to do something! Further, if the call did not come from a person with appropriate authority, why should one obey it? To use an earlier example, if Socrates had come to believe that the prophetess was caused to say "Know thyself!" by volcanic fumes, he could not claim to know that the life of self-examination was the true calling of a philosopher. The words "Know thyself!" would be empty sounds, not a calling from someone who had the authority to issue it.

## 6. Conclusions

In this paper, I tried to do four main things. First, I attempted to make it plausible that conscience is a faculty capable of moral self-knowledge, siding broadly with moral realism and against anti-realist approaches. Second, I analyzed the nature of moral obligations, claiming that they are naturally received as commands (and hence are inherently meaningful), they are prescriptive, and personal, and that they possess a kind of universal, normative necessity. Third, I argued that materialism cannot explain the conscience (as I defined it) because it fails to ground the existence of moral obligations and does not account for our knowledge of them even if they do exist. Finally, I argued that Christian theism provides a more plausible foundation for the existence and knowledge of moral obligations, and hence provides a better explanation of the conscience. This argument parallels a version of the argument from reason that sees some version of theism as superior to materialism in its ability to explain our capacity to reason in accordance with the laws of Logic.

**Funding:** This research received no external funding.

**Data Availability Statement:** Not applicable.

**Acknowledgments:** Many thanks to three peer reviewers and the editors of this special issue for their insightful comments on earlier versions of this paper. Reflection on these comments greatly improved this paper.

**Conflicts of Interest:** The author declares no conflict of interest.

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
