# Peer review of "Grounding the Conscience"

_religions, doi:10.3390/rel13100966_

Round 1
Reviewer 1 Report
Overall, this is a clearly written essay that does a good job of accomplishing its objective. Though the author explores several major sub points, the author still does a good job of leading the reader through them. Overall, I think it is well argued & addresses the key points of the issue involved. Still, I do have some suggestions for revisions:
1. Starting on p. 2, lines 46-47, the author begins to see a key connection between moral commands and language. This occurs again on p. 6, lines 255-56. More so, the author suggests that even thought itself could be linguistic (263; but, please compare Dallas Willard, “The Absurdity of Thinking in Language”). Indeed, this seems to be a claim that is reiterated in several places.
Yet, on p. 8, the author says “it seems that information, in the form of instructions, is in general 378 something more than matter itself” (lines 378-79). This seems to be a nod to propositions as distinct ontologically from sentences. There seems to be a similar kind of claim on p. 13, lines 632-33: “We saw above several reasons for thinking that moral commands, the natural expression of moral obligations, are immaterial entities.”
The key issue is: what kind of thing are sentences? In agreement with JP Moreland, it seems to me that they are sense perceptible, physical things, yet they can be used to express an underlying meaning (i.e., a proposition). I think the author should carefully consider the strong emphasis upon moral obligations, thoughts, etc. as linguistic, for that seems to play into the materialist’s hand. Yet, quite arguably, these things themselves are not; but, they can be expressed by sentences that are physical.
I think on p. 12, this close association of language & reality could get the author’s argument into trouble. Why think that reality is linguistic, even from the reasons given in that context? Are reasons linguistic? Again, they can be expressed in language, but it seems propositions are what are being so expressed. Is the Logos linguistic?
2. An observation: 3.4. Moral Obligations Express Universal, Normative Necessity: of course, postmoderns will deny this. Would it help to speak to such mindsets too? Or would you (the author) want to stick mainly with materialists? (Of course, there are several Christians who have made the postmodern turn & are physicalists.)
3. Section 4.1: I think there is another kind of move a materialist could make, like Christine Korsgaard: we impose our form (obligations) upon matter, to yield normative, universal obligations (like Kant). I think it would be worthwhile to address her kind of view.
4. P. 12, lines 568-69: is intentionality, the ofness or aboutness of almost all mental states, a power? Or is it a property? I think we have the power (ability) to form & even act on intentions (purposes), but those of course are not identical with intentionality.
I think my points 1 & 3 are most important.
Author Response
Very helpful, thank-you!
Please see the attachment for a point-by-point response.

Reviewer 2 Report
I can say the article is qualified although I am disturbed at times by some transitions (f. e., from the argumentative level to references from biblical literature, which materialists wouldn't accept).
1. What the author is trying to demonstrate is the immateriality of our moral obligations and not their christian basis, or? He writes:
Rather, as I argue in section 5, the conscience is better explained by Christian theism. This, I maintain, is because moral obligations are naturally received as commands of practical reason that are knowable only if a divine personal Logos governs the world...
I haven't found a relevant argument that demonstrates why Christian theism is the only right one. Instead, there are quite a few quotes that mix the ancient Socrate's daimonion message with the voice of conscience. Does it mean - there is no difference? Between ancient and catholic Bible?
2. Citation: "So when we wrong non-persons, it is at least arguable that this is ultimately because there is a person whom is wronged...." Really? what if the dog doesn't belong to anyone? It means, It isn't bad action to kill him? Is our judgeing all person-centered?
Author Response
Thank-you for the comments!
The attached contains my point-by-point response.

Reviewer 3 Report
I really appreciated this paper and found its general argument to be convincing. I found its analysis of the insufficiency of materialism as a ground for the obligations of conscience to be more persuasive than its analysis of the sufficiency of Christian theism as a ground for conscience (though I still found the argument about Christian theism to be persuasive). One point in the first part of the paper that would be good to clarify is the appeal to language. I found this confusing and not fully explained. In a way, I found the reference to language raised more questions than it answered (e.g., is language a material reality and, if not, why not?). Could the argument be just as strong without bringing language into the article? Or could there be a clearer justification for what language adds to the argument? I think I found the discussion of Christian theism less persuasive than the first part of the paper because the more distinct aspects of the Christian God - a Trinitarian character; the Incarnation; the Cross - didn't end up playing too much of a role in the argument as compared to the aspects of the Christian God that are shared with other Abrahamic traditions. Also, I wondered if there might be some way to appeal in this section to a Christian philosopher like Plantinga. As I read the paper, I found myself wondering if he had discussed things relevant to the argument. Also, Newman's thought on conscience might be helpful to the argument. He locates belief in God in the experience of conscience itself (so not so much in something like the Five Ways). Could Newman's arguments in this manner help clinch the point of the paper?
Author Response
Thank-you for the helpful comments!
Please see the attached response.
